# Long Working Hours Indirectly Affect Psychosomatic Stress Responses via Complete Mediation by Irregular Mealtimes and Shortened Sleep Duration: A Cross-Sectional Study

**DOI:** 10.3390/ijerph19116715

**Published:** 2022-05-31

**Authors:** Tenshi Watanabe, Jiro Masuya, Shogo Hashimoto, Mina Honyashiki, Miki Ono, Yu Tamada, Yota Fujimura, Takeshi Inoue, Akiyoshi Shimura

**Affiliations:** 1Department of Psychiatry, Tokyo Medical University, 6-7-1 Nishi-Shinjuku, Shinjukuku 160-0023, Tokyo, Japan; m73n2d@gmail.com (T.W.); j-masuya@tokyo-med.ac.jp (J.M.); honyashiki@outlook.jp (M.H.); mikky@tokyo-med.ac.jp (M.O.); y-tmd@umin.ac.jp (Y.T.); fyota@yahoo.co.jp (Y.F.); tinoue@tokyo-med.ac.jp (T.I.); 2Department of Anesthesiology, Ebara Hospital, 4-5-19 Higashi-Yukigaya, Otaku 145-0065, Tokyo, Japan; 3Department of Psychiatry, Hokkaido University Graduate School of Medicine, Kita 10, Nishi 7, Kitaku, Sapporo 060-0808, Hokkaido, Japan; hassy19710310@gmail.com; 4Department of Psychiatry, Toranomon Hospital Kajigaya, 1-3-1 Kajigaya, Kawasaki 213-8587, Kanagawa, Japan; 5Department of Psychiatry, Tokyo Medical University Hachioji Medical Center, 1163 Tatemachi, Hachiojishi 193-0998, Tokyo, Japan

**Keywords:** overtime work, sleep, mealtimes, psychosomatic stress responses, mediator

## Abstract

Background: Long working hours are detrimental to physical and mental health. However, the association between long working hours and psychosomatic symptoms have remained controversial, possibly because of the existence of mediators between working hours and psychosomatic stress responses. We hypothesized that lifestyle habits, regarding sleep and mealtimes, act as mediators, and analyzed the associations between long working hours, sleep duration, mealtime regularity, and psychosomatic stress responses in office workers. Methods: From April 2017 to March 2018, an online cross-sectional survey regarding overtime work hours, work-related stress, sleep, and eating habits was conducted with employees of 17 companies located in Tokyo, Japan. Answers were obtained from 3559 employees, and 3100 provided written consent for the academic use of their answers, and were included in the analysis. A path analysis was conducted to assess the effect of overtime work on psychosomatic stress via shortened sleep or irregular mealtimes. Results: Overtime work hours had no direct effect on psychosomatic stress responses and depressive symptoms. However, overtime work hours affected sleep duration and the regularity of mealtimes. The effects of overtime work hours on psychosomatic stress responses and depressive symptoms were completely mediated by sleep duration and the regularity of mealtimes. Conclusion: Long working hours do not affect mental health directly; however, shortened sleep duration and irregular mealtimes mediate the effect of long working hours on psychosomatic stress responses and depressive symptoms.

## 1. Introduction

Occupational stress is known to have multifarious detrimental effects on physical and mental health, and has recently been gaining attention. The National Institute for Occupational Safety and Health has defined occupational stress as harmful physical and emotional responses that occur when the requirements of a job do not match the capabilities, resources, or needs of the worker, and warned that it can lead to poor health and injuries. Physically, occupational stress is a risk factor for ischemic heart disease [1], similarly to psychological distress [2] and perceived mental stress [3], being linked to coronary heart disease. Occupational stress is also associated with obesity [4] and impaired glucose tolerance [5]. Regarding mental health, although it has been under debate, occupational stress appears to play a role in the development of depressive disorders [6].

The word “Karoshi” is a loanword from Japanese, meaning death as a result of extremely long overtime work hours. Long overtime work hours are known to be an independent risk factor of fatal cardiovascular events [7,8]. Although it has also been suggested that overtime work leads to occupational injuries [9], the association was shown to be ambiguous or weak [10]. However, the association between a worker’s mental health state and the length of overtime work has not been clarified to date [11,12]. A previous study indicated the association between overtime work hours and occupational stress [8], but its actual effect on mental disorders, such as depressive and anxiety disorders, remains controversial. A meta-analysis suggested an association between long overtime work hours and depressive symptoms [11]. However, another meta-analysis concluded that the association was “small if not negligible” [12]. Moreover, a cohort/case-control study on Japanese office workers found no significant association between overtime work hours and the onset of depression [13].

From the viewpoint of public health, the improvement of mental health in the workplace is a matter of great importance. To achieve this, policy-making aimed to regulate and reduce overtime work hours has been performed in many countries. However, if the effects of overtime work hours on mental health are not of clinical significance, the benefits of such policies will become questionable. For example, data published by the Japanese Business Federation showed a steady downtrend from 2017 to 2019 in the total working hours of workers of all sectors, which the federation attributed to the enactment of workstyle reform laws [14]. In contrast, the number of claims and payments for mental disorders certified as industrial accidents showed no improvement during the same period [15]. To evaluate this situation, the effect of overtime work hours on mental health should be investigated, and if it is significant, its pathway should be clarified. To our knowledge, studies to achieve this goal have not been conducted to date.

These inconsistent and ambiguous results of the association between overtime work and mental health from previous studies suggest the existence of yet-unidentified mediators. Sleep is a candidate of such a mediator, as it is an important aspect of life, and is vital for maintaining physical and mental health and daytime function. In addition, sleep is known to be affected by overtime work hours [16,17,18], and could possibly be shortened by long overtime work hours [18]. Dietary habits are another possible factor, as they play an important role in maintaining health. Personal preferences will likely have an effect on the contents of one’s diet, but it can be deduced that long overtime work hours may disrupt the rhythm of mealtimes, which may lead to detrimental effects on mental health [18,19,20].

Therefore, in this study, we aimed to clarify the hypothesis that (1) the effect of overtime work on mental health is mediated by specific factors, and (2) lifestyle habits, which are affected by long work hours, such as sleep and the regularity of mealtimes, play a role as mediators.

## 2. Methods

This study was performed as an online cross-sectional survey asking about work-time, work-associated stress, sleep, and eating habits, and was conducted from April 2017 to March 2018. A total of 17 companies located in Tokyo, Japan, participated in this survey. The study group consisted of white-collar companies, such as those working in information technology, game development, finance, broadcasting, consulting, and trading, and a governmental public office, travel agency, patent agency, and temp agency. As inclusion criteria, all of the staff in the companies, including the part-time workers, worked for more than 30 h per week. There were no exclusion criteria. Answers were obtained from 3559 employees from these companies, and a total of 3100 employees provided written informed consent for their answers to be used for academic research, and were included in the analysis. As answering all questions were set as being mandatory, there were no missing answers. This study was conducted under the approval of the Ethics Committee of Tokyo Medical University (study approval no.: SH3652).

The survey was self-administered, and consisted of the Brief Job Stress Questionnaire (BJSQ) [21], the Pittsburgh Sleep Quality Index (PSQI) [22], and original questions regarding the respondents’ demographics, length of overtime work hours per month, and meal pattern.

BJSQ is a questionnaire consisting of 57 Likert scale questions designed to assess job stressors, the severity of psychosomatic symptoms occurring as stress responses, and social factors that modify the association between the two factors. The psychosomatic symptoms score of the BJSQ indicates the respondent’s psychological and physical stress levels, and can be subdivided into the following six categories: lack of liveliness, irritability, fatigue, anxiety, depression, and somatic symptoms. BJSQ is authorized by the Japanese Ministry of Health, Labor and Welfare as the standard method for measuring occupational stress in Japan.

PSQI is a widely used and validated method for assessing sleep disturbances, both quantitatively and qualitatively. PSQI is comprised of seven components, designed to assess a respondent’s sleep, namely, sleep quality, sleep latency, sleep duration, sleep efficiency, night time sleep disturbance, use of sleep medication, and daytime dysfunction.

Overtime work hours were defined as the per month total working hours exceeding 40 h per week, according to the definition of the Labor Standards Act of Japan.

The regularity of mealtimes was evaluated by the subjective four-point Likert scale, “Quite regular, tend to be regular, tend to be irregular, irregular”, in which the former two were classified as “regular”, and the latter two as “irregular” in this analysis. In addition to the regularity of mealtimes, the frequency of vegetable intake, which may also be associated with mental health, was evaluated by the subjective four-point Likert scale, “every meal, every day, less than every day, almost none”.

The associations between each of the variables were evaluated by correlation analyses and covariance structure analyses to analyze the mediating effects of the variables. SPSS ver. 28 (IBM, Armonk, NY, USA) and Mplus ver. 8.5 software (Muthén & Muthén, Los Angeles, CA, USA) were used for the statistical analyses.

## 3. Results

Of the 3100 participants, 1927 (62.1%) were men, the mean age was 36.6 years (±9.3 years), and the mean overtime work hours was 24.4 h (±27.3 h), with 433 (14.0%) participants performing more than 45 h of overtime work per month. A total of 1849 (59.6%) participants reported regular mealtimes, and 1938 (62.5%) reported a regular vegetable intake habit (Table 1). Participants with long overtime work showed a tendency towards irregular mealtimes (*T* = 9.67, *p* < 0.001), but no significant decrease in the intake of vegetables (*T* = 1.13, *p* = 0.258).

Simple correlation analysis between overtime work hours and psychosomatic stress responses, which includes depression and fatigue sub-scores, showed a weak positive correlation (*r* = 0.060, *p* = 0.001); however, there was no significant correlation between overtime work hours and depression sub-score (*r* = 0.010, *p* = 0.593) (Table 2).

Figure 1 shows the structural equation model based on our hypothesis that overtime work tends to result in irregular mealtimes, and also shortens sleep duration, and through these factors, overtime work affects psychosomatic stress responses. This was a saturated model. Regarding the direct effect of overtime work hours on psychosomatic stress responses, the effect was insignificant with a standardized path coefficient of −0.003 (*p* = 0.871). However, the paths leading to the psychosomatic stress responses from irregular mealtimes and total sleep time were both statistically significant, with standardized path coefficients of 0.239 (*p* = 0.000) and −0.145 (*p* = 0.000), respectively.

The indirect effects of overtime work hours on the psychosomatic stress responses are shown in Table 3. The effect was significantly mediated by irregular mealtimes and total sleep time. Thus, the effect of overtime work hours on the psychosomatic stress responses were found to occur completely via this mediation effect. The total standardized path coefficient of the indirect effect was 0.063, and was statistically significant (*p* < 0.001). We also found that this model explained 9.0% of the change in psychosomatic stress caused by overtime work hours, according to the R-squared value.

## 4. Discussion

In this study, through covariance structure analysis, we showed that overtime work hours indirectly affect psychosomatic stress responses in adult workers, and these effects are completely mediated by two factors; i.e., shortened sleep duration and irregular mealtimes.

Previous studies have aimed to investigate the association between various physical and mental disorders and overtime work hours, but the results have been controversial [8,11,12]. We believe that the two mediators we identified in this study may explain the inconsistencies in the literature regarding overtime work hours and psychosomatic stress responses, which are predisposing factors to various medical conditions. Furthermore, our findings are expected to lead to a better understanding of mental health and methods for its improvement in the workplace. In this study, a correlation between overtime work hours and psychosomatic stress responses was detected; however, it was very weak, and a significant correlation was not detected between overtime work hours and depressive symptoms, while a previous study even showed a link between overtime work hours and increased vigorousness in male workers [23]. Hence, we believe that simply focusing only on the length of work hours and reducing it is insufficient for the maintenance of mental health in workplaces, and that further measures towards maintaining appropriate sleeping and eating habits are necessary.

As it should be possible for people to maintain well-regulated eating habits, or achieve sufficient total sleep time even with a few extra hours of work in a day, long working hours do not directly result in shortened sleep time or irregular mealtimes. It is also likely that there is a high variability among individuals in the ways overtime work hours affect eating and sleeping habits, making the direct effect of overtime work on psychosomatic stress responses and depression weak or insignificant. It can also be argued that irregular mealtimes and shortened sleep time may be a result of a multitude of factors in addition to long working hours. Mealtime regularity, which is included in the Social Rhythm Metric, is a factor that affects mental health [19]. However, mealtimes, as is the case with other activities that are performed daily, are not naturally consistently regular, and require the effort of the individual to keep them regular. Furthermore, sleep duration has been shown to be shortened by factors outside of the workplace, such as housework, childcare [24], and commuting [25,26]. Hence, we can argue that the existence of multiple factors other than those of the workplace that have an effect on sleep duration and irregular mealtimes adds to the inconsistency of the effect of overtime work hours on mental health. It is also important to note that regarding sleep, types of disturbances other than short sleep duration substantially affect mental health.

Specific stress factors in the workplace that have been shown to lead to sleep disturbance include effort-reward imbalance [27], ruminations of adverse workplace experiences [28], and prolonged harassment (bullying) [29]. On the other hand, it has been shown that higher job control and social support may alleviate sleep disturbances [30]. In light of this, we believe that maintaining a civil workplace, where workers receive fair treatment, have more control over their jobs, and are exposed to less negative experiences, is generally beneficial for workers’ mental health, in addition to the shortening of work hours.

It is also noteworthy that sleep disturbance is deeply associated with people’s lifestyles, making it a condition that is relatively easy for people to alleviate or aggravate on their own. Some known lifestyle habits that have been shown to be detrimental to sleep include the use of electric displays immediately before sleep [31], alcohol/caffeine/nicotine intake prior to sleep [32,33], and some diet-associated traits, such as low vegetable consumption, increased snacking, and frequently skipping breakfast [20]. In our study, low vegetable consumption had a significant but weak association with psychosomatic stress responses. However, as the effect of irregular mealtimes was more prominent, and also more likely to be affected by overtime work hours, irregular mealtimes are expected to play a more substantial role than vegetable intake in the effect of overtime work hours on psychosomatic stress responses.

Through our findings, we demonstrated that improving mental health in the workplace requires a holistic approach with an emphasis on sleep and eating habits, rather than simply shortening working hours. Although disruptions in lifestyle habits may have their origins in the workplace, there appear to be many habits in present workers’ lifestyles that can be improved. To our knowledge, this is the first study to suggest an explanation for the inconsistent or weak results of previous studies focusing on the association between long overtime work hours and mental health, and moreover, this is the first report demonstrating that sleep and meal play key mediating roles between them. These results may be useful for performing more effective and specific health guidance in the future.

This study has some potential limitations. Firstly, the data obtained from our study population were all from questionnaires, in which the participants provided answers regarding both exposure and outcome. Although the valid response rate remained high at 87.1%, the nature of questionnaires limits the reliability and validity of the obtained data, making our study prone to self-report bias, recall bias, and spurious correlations. Secondly, as this study implemented a cross-sectional study design, there is no evidence of a temporal association between the exposure (disrupted sleep and eating) and outcome (increased occupational stress). Thirdly, the Likert scales used in our study to assess the regularity of mealtimes and vegetable intake were our own original scales, and have not been validated. Fourthly, all of the participants enrolled in this study worked in offices located in Tokyo, and voluntarily participated in our study, and may hence have higher motivation towards health promotion than workers who did not participate. Therefore, our results may not apply to working populations in different settings. Lastly, other important controlling variables, such as smoking status, and psychological baselines, such as neuroticism, were not surveyed, and hence not adjusted for. Accordingly, a longitudinal study on a wider range of workers should be considered in the future.

## 5. Conclusions

Long working hours did not affect psychosomatic stress responses directly; however, irregular mealtimes and shortened sleep duration mediated the effects of long working hours on psychosomatic stress responses. As the shortening of overtime work hours does not directly reduce psychosomatic stress responses, getting enough hours of sleep and regular mealtimes are important for maintaining workers’ mental health. Furthermore, even if working for long hours, taking measures towards preserving sleep duration and mealtime regularity may prevent psychosomatic stress responses.

## Figures and Tables

**Figure 1 ijerph-19-06715-f001:**
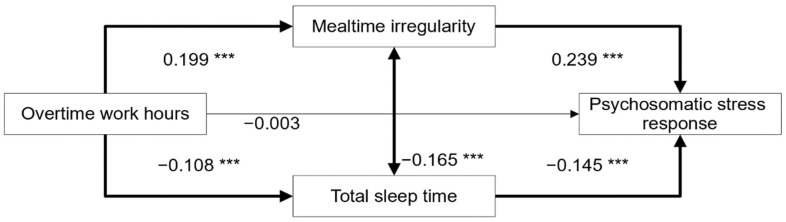
Structural equation model of the effect of overtime work hours on psychosomatic stress responses. The bold arrows represent statistically significant direct paths. The numbers adjacent to the arrows indicate the standardized path coefficients. *** *p* < 0.001.

**Table 1 ijerph-19-06715-t001:** Demographics of the respondents.

		Depression	Psychosomatic Stress Responses
	n (%)	Mean (SD)	Mean (SD)
Total sample	3100 (100%)	10.12 (3.60)	56.7 (13.9)
Overtime work			
<45 h/week	2667 (86.0%)	10.08 (3.58)	56.3 (13.8)
≥45 h/week	433 (14.0%)	10.33 (3.76)	59.3 (14.4)
*T*-value	-	1.36 (*p* = 0.172)	4.14 (*p* < 0.001)
Sex			
Male	1972 (62.2%)	9.92 (3.48)	55.4 (13.4)
Female	1167 (37.6%)	10.43 (3.77)	58.8 (14.4)
*T*-value	-	3.74 (*p* < 0.001)	6.43 (*p* < 0.001)
Mealtime regularity			
Regular	1849 (59.6%)	9.48 (3.30)	53.7 (12.8)
Irregular	1251 (40.4%)	11.05 (3.82)	61.2 (14.3)
*T*-value	-	11.86 (*p* < 0.001)	15.0 (*p* < 0.001)
Vegetable intake			
Every day	1938 (62.5%)	9.68 (3.43)	54.8 (13.4)
Less than every day	1162 (37.5%)	10.84 (3.76)	59.9 (14.3)
*T*-value	-	8.58 (*p* < 0.001)	9.84 (*p* < 0.001)

**Table 2 ijerph-19-06715-t002:** Correlation between overtime work hours, psychosomatic stress responses, and depression.

	Pearson’s *r*-Coefficient		
	Overtime Work Hours	Psychosomatic Stress Responses	Depression
Overtime work hours	1	0.060 (*p* = 0.001)	0.010 (*p* = 0.593)
Psychosomatic stress responses		1	0.857 (*p* < 0.001)
Depression			1

**Table 3 ijerph-19-06715-t003:** Standardized path coefficients between variables.

	Direct Effect to		
From	Mealtime Regularity	Total Sleep Time	Psychosomatic Stress Responses
Overtime work hours	0.199 ***	−0.108 ***	−0.003
Mealtime irregularity	-	−0.165 ***	0.239 ***
Total sleep time			−0.145 ***
	Indirect effect to		
From	Via		Psychosomatic stress responses
Overtime work hours	Mealtime regularity		0.048 ***
	Total sleep time		0.016 ***
	Total effect to		
From			Psychosomatic stress responses
Overtime work hours			0.060 **

** *p* < 0.01, *** *p* < 0.001.

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
