# Peer review of "Long Working Hours Indirectly Affect Psychosomatic Stress Responses via Complete Mediation by Irregular Mealtimes and Shortened Sleep Duration: A Cross-Sectional Study"

_ijerph, 2022, doi:10.3390/ijerph19116715_

Round 1
Reviewer 1 Report
The article deals with a very interesting but also important issue regarding long working hours in terms of stress. The abstract is well-described and substantive. The research part lacks information on how the respondents were selected. I propose to improve the work with these aspects: who was the target group; was gender, age, or the number of hours taken into account? There is also no information on where the survey was posted or whether it was a CAWI survey. The introduction lacks the hypotheses or the goals of the study. Authors should also make clear what is new in their work, which makes it different from other works on the same subject. The strength of the work is the topic and analysis, while the weakness is the review of the literature as well as conclusion, which requires rebuilding.
Author Response
Dear Reviewers,
We thank you very much for reviewing our paper and for your valuable and insightful comments, which have greatly helped us improve the quality of our manuscript. We have carefully considered the comments, and have tried our best to address every one of them. Our point-by-point responses to the comments are shown below. We hope that the revised manuscript is now acceptable for publication in International Journal of Environmental Research and Public Health.
Comments of Reviewer 1
Comment #1:
The article deals with a very interesting but also important issue regarding long working hours in terms of stress. The abstract is well-described and substantive. The research part lacks information on how the respondents were selected. I propose to improve the work with these aspects: who was the target group; was gender, age, or the number of hours taken into account? There is also no information on where the survey was posted or whether it was a CAWI survey.
Response:
We thank you for the valuable suggestions. To make the study sample clear, we added the following description to the revised manuscript. The survey was performed via the internet but was self-administered, and was not in an interview form, and this was also described in the revised manuscript, as follows.
Abstract (lines 20–22)
“Methods: From April 2017 to March 2018, an online cross-sectional survey regarding overtime work hours, work-related stress, sleep, and eating habits was conducted on 17 companies located in Tokyo, Japan”
Methods (lines 87-99)
“This study was performed as an online cross-sectional survey asking about work-time, work-associated stress, sleep, and eating habits, and was conducted from April 2017 to March 2018. A total of 17 companies located in Tokyo, Japan, participated in this survey. The study group consisted of white-collar companies, such as those working in information technology, game development, finance, broadcasting, consulting, and trading, and a governmental public office, travel agency, patent agency, and temp agency. As inclusion criteria, all of the staff in the companies, including the part-time workers, worked for more than 30 hours per week. There were no exclusion criteria. Answers were obtained from 3,559 employees from these companies, and a total of 3,100 employees provided written informed consent for their answers to be used for academic research, and were included in the analysis. As answering all questions were set as being mandatory, there were no missing answers. This study was conducted under the approval of the Ethics Committee of Tokyo Medical University (study approval no.: SH3652).”
Comment #2
The introduction lacks the hypotheses or the goals of the study
Response:
Thank you for the comment. We have modified the Introduction section to clearly state the hypothesis in the revised manuscript, as follows.
Introduction (lines 82–85)
“Therefore, in this study, we aimed to clarify the hypothesis that (1) the effect of overtime work on mental health is mediated by specific factors, and (2) lifestyle habits, which are affected by long work hours, such as sleep and the regularity of mealtimes, play a role as mediators.”
Comment #3
Authors should also make clear what is new in their work, which makes it different from other works on the same subject.
Response:
Thank you for the valuable comment. To emphasize the new points of our present study, we have added the following text to the Discussion section of our revised manuscript.
Discussion (lines 223–227)
“To our knowledge, this is the first study to suggest an explanation for the inconsistent or weak results of previous studies focusing on the association between longtime overwork and mental health, and moreover, this is the first report demonstrating that sleep and mealtime irregularities play key mediating roles between them. These results may be useful for performing more effective and specific health guidance in the future.”
Comment #4
Extensive editing of English language and style required
Response:
We apologize for our non-well-structured English descriptions. The English has been proofread by a native English speaker with a knowledge of science. We hope that the language in the manuscript is now suitable for publication.
Reviewer 2 Report
ijerph-1628630
Long Working Hours Indirectly Affects Psychosomatic Stress Responses, Completely Mediated by Irregular Mealtimes and Shortened Sleep Duration
This article focuses on how dietary patterns and sleep disturbances could act as mediators on the psychosomatic stress response related to extended working hours.
The present paper is interesting, is well organised and well written.
The title and the abstract of the paper should clearly indicate the design of the study, in this case a cross-sectional study.
In the Introduction I think the notion “stress” should be better defined. Are the authors referring to “distress”? Please take look at references such as Iso H, Circulation 2002 and Stansfeld SA, 2002.
As to overtime work the literature is conflicting. Please look at Matre D, 2021.
The hypothesis and the aim of the study in the introduction is not well organised. For example, one hypothesis appears in line 64-66 and the other one in line 77-80. In line 81-82 a statistical approach is mentioned, this should be moved to the Methods part.
My main concern of the paper is the methods used. The authors point out a limitation namely its cross-sectional design. However, I believe another methodological weakness could be worse. One questionnaire is used, and this creates major methodological problems, serious bias which occurs when the participants provide answers on both exposure and outcome. This risk of self-report bias could lead to an ‘artifactual covariance between the predictor and criterion variable’ since the same person is assessing both measures.
The authors do not provide any inclusion or exclusion criteria of the group participated in the study. How were the participants selected? Any control subjects? How was the study size arrived at?
The statistical part should be extended. The model used in figure 1 should be explained. Also, how did the authors address missing data?
The results. Did the authors consider using a flow diagram? There is a lack of demographic data, for instance smoking habits, neuroticism. Confounders adjusted for?
The discussion on the possibly strengths and limitations of the study should be extended.
Minor comments:
Linguistically the paper is good, but improvements could be necessary.
“Fiscal year” sounds weird- a better expression?
Do not use “gender” use “sex”.
The results and the analysis in the paper is interesting to pass on to the scientific community along with suggestions of how to improve the conditions for workers by enforcing control of extended working hours.
Author Response
Dear Reviewers,
We thank you very much for reviewing our paper and for your valuable and insightful comments, which have greatly helped us improve the quality of our manuscript. We have carefully considered the comments, and have tried our best to address every one of them. Our point-by-point responses to the comments are shown below. We hope that the revised manuscript is now acceptable for publication in International Journal of Environmental Research and Public Health.
Comments of Reviewer 2
Comment #1
This article focuses on how dietary patterns and sleep disturbances could act as mediators on the psychosomatic stress response related to extended working hours.
The present paper is interesting, is well organised and well written. The title and the abstract of the paper should clearly indicate the design of the study, in this case a cross-sectional study.
Response:
We appreciate your positive evaluation of our manuscript, and also thank you for your valuable suggestion. We have added the study design to our title, which is now “Long Working Hours Indirectly Affects Psychosomatic Stress Responses via Complete Mediation by Irregular Mealtimes and Shortened Sleep Duration: A Cross-Sectional Study,” and have also modified our abstract, as follows.
Revised manuscript:
Abstract (lines 20–23)
“Methods: From April 2017 to March 2018, an online cross-sectional survey regarding overtime work hours, work-related stress, sleep, and eating habits was conducted on 17 companies located in Tokyo, Japan.”
Methods (lines 87-89)
“This study was performed as an online cross-sectional survey asking about work-time, work-associated stress, sleep, and eating habits, and was conducted from April 2017 to March 2018.”
Comment #2
In the Introduction I think the notion “stress” should be better defined. Are the authors referring to “distress”? Please take look at references such as Iso H, Circulation 2002 and Stansfeld SA, 2002.
Response:
Thank you for pointing this out. We added the definition of “occupational stress” as described by the National Institute for Occupational Safety and Health, and also referred to the citations suggested by the reviewer in the revised manuscript, as follows.
Introduction (lines 37-44)
“The National Institute for Occupational Safety and Health defined occupational stress as harmful physical and emotional responses that occur when the requirements of a job do not match the capabilities, resources, or needs of the worker, and warned that it can lead to poor health and even injury. Physically, occupational stress is a risk factor for ischemic heart disease [1], similarly to psychological distress [2] and perceived mental stress [3] being linked to coronary heart disease. Occupational stress is also associated with obesity [4], and impaired glucose tolerance [5].”
Added citations
Stansfeld, Stephen A., et al. "Psychological distress as a risk factor for coronary heart disease in the Whitehall II Study." International journal of epidemiology 31.1 (2002): 248-255.
Iso, Hiroyasu, et al. "Perceived mental stress and mortality from cardiovascular disease among Japanese men and women: the Japan Collaborative Cohort Study for Evaluation of Cancer Risk Sponsored by Monbusho (JACC Study)." Circulation 106.10 (2002): 1229-1236.
Comment #3
As to overtime work the literature is conflicting. Please look at Matre D, 2021.
Response:
Thank you for the advice and introducing us to important literature that we have missed. We have read thet papers and recognized that there is also a weak or ambiguous association between overtime work and incident risk. We have added this to the Introduction section of our revised manuscript, together with aand citation, as follows.
Introduction (lines 47-50)
“Long overtime work hours are known to be an independent risk factor of fatal cardiovascular events [7,8]. Although it has also been suggested that overtime work leads to occupational injuries [9], the association was shown to be ambiguous or weak [10].”
Added citation
Matre, Dagfinn, et al. "Safety incidents associated with extended working hours. A systematic review and meta-analysis." Scandinavian journal of work, environment & health 47.6 (2021): 415.
Comment #4
The hypothesis and the aim of the study in the introduction is not well organised. For example, one hypothesis appears in line 64-66 and the other one in line 77-80. In line 81-82 a statistical approach is mentioned, this should be moved to the Methods part.
Response:
Thank you for the valuable suggestions. We have reorganized the explanation of our hypothesis in the revised manuscript as follows, to make our manuscript easier to understand.
Introduction (lines 72-74)
“These inconsistent and ambiguous results of the association between overtime work and mental health from previous studies suggest the existence of yet-unidentified mediators.”
Introduction (lines 82-85)
“Therefore, in this study, we aimed to clarify the hypothesis that (1) the effect of overtime work on mental health is mediated by specific factors, and (2) lifestyle habits, which are affected by long work hours, such as sleep and the regularity of mealtimes, play a role as mediators.”
Methods (lines 124-126)
“The associations between each of the variables were evaluated by correlation analyses and covariance structure analyses to analyze the mediating effects of the variables.”
Comment #5
My main concern of the paper is the methods used. The authors point out a limitation namely its cross-sectional design. However, I believe another methodological weakness could be worse. One questionnaire is used, and this creates major methodological problems, serious bias which occurs when the participants provide answers on both exposure and outcome. This risk of self-report bias could lead to an ‘artifactual covariance between the predictor and criterion variable’ since the same person is assessing both measures.
Response:
Thank you for this important point. We have extended our study limitations in the Discussion section of our revised manuscript, as follows.
Discussion (lines 229-223)
“This study has some potential limitations. Firstly, the data obtained from our study population were all from questionnaires, in which the participants provided answers regarding both exposure and outcome. Although the valid response rate remained high at 87.1%, the nature of questionnaires limits the reliability and validity of the obtained data, making our study prone to self-report bias, recall bias, and make spurious correlations.”
Comment #6
The authors do not provide any inclusion or exclusion criteria of the group participated in the study. How were the participants selected? Any control subjects? How was the study size arrived at?
Response:
Thank you for the important suggestions. We have added a detailed description about the participants to the revised manuscript, as follows. There are no control subjects in this study.
Methods (lines 87-94)
“This study was performed as an online cross-sectional survey asking about work-time, work-associated stress, sleep, and eating habits, and was conducted from April 2017 to March 2018. A total of 17 companies located in Tokyo, Japan, participated in this survey. The study group consisted of white-collar companies, such as those working in information technology, game development, finance, broadcasting, consulting, and trading, and a governmental public office, travel agency, patent agency, and temp agency. As inclusion criteria, all of the staff in the companies, including the part-time workers, worked for more than 30 hours per week. There were no exclusion criteria.”
Discussion (lines 238-241)
“Fourthly, all of the workers enrolled in this study worked in offices located in Tokyo, and voluntarily participated in our study, and may hence have higher motivation towards health promotion than workers who did not participate. Therefore, our results may not translate well to the working population in different settings.”
Comment #7
The statistical part should be extended. The model used in figure 1 should be explained. Also, how did the authors address missing data?
Response:
We thank you for mentioning it. We have added the explanations, as follows.
Methods (lines 104-105)
“As answering all questions were set as being mandatory, there were no missing answers.”
Results (lines 97-98)
“Figure 1 shows the structural equation model based on our hypothesis that overtime work tends to result in irregular mealtimes, and also shortens sleep duration, and through these factors, overtime work affects the psychosomatic stress responses. This was a saturated model. ”
Comment #8
The results. Did the authors consider using a flow diagram? There is a lack of demographic data, for instance smoking habits, neuroticism. Confounders adjusted for?
Response:
We thank you for the advice. As described above (answers to comments #6 and #7), there was only one simple criterion and no exclusion criteria. We are sorry that we did not analyze the status of the participants, such as smoking habits and neuroticism, and hence could not adjust them. We have also added this to our limitations, as follows.
Discussion (lines 241-243)
“Lastly, other important controlling variables, such as smoking status, or psychological baselines, such as neuroticism were not surveyed, and not adjusted.”
Comment #9
The discussion on the possibly strengths and limitations of the study should be extended.
Response:
Thank you for the advice. We have now included a full description of the possible strengths and limitations of the study, as follows.
Discussion (lines 223-244)
“To our knowledge, this is the first study to suggest an explanation for the inconsistent or weak results of previous studies focusing on the association between longtime overwork and mental health, and moreover, this is the first report demonstrating that sleep and mealtime irregularities play key mediating roles between them. These results may be useful for performing more effective and specific health guidance in the future.
This study has some potential limitations. Firstly, the data obtained from our study population were all from questionnaires, in which the participants provided answers regarding both exposure and outcome. Although the valid response rate remained high at 87.1%, the nature of questionnaires limits the reliability and validity of the obtained data, making our study prone to self-report bias, recall bias, and make spurious correlations. Secondly, as this study implemented a cross-sectional study design, there is no evidence of a temporal association between the exposure (disrupted sleep and eating) and outcome (increased occupational stress). Thirdly, the Likert scale used in our study to assess mealtime irregularities and vegetable intake were our own original scale, and therefore not validated. Fourthly, all of the workers enrolled in this study worked in offices located in Tokyo, and voluntarily participated in our study, and may hence have higher motivation towards health promotion than workers who did not participate. Therefore, our results may not translate well to the working population in different settings. Lastly, other important controlling variables, such as smoking status, or psychological baselines, such as neuroticism were not surveyed, and not adjusted. Accordingly, a longitudinal study based on a wider target should be considered in the future.”
Comment #10
Minor comments:
Linguistically the paper is good, but improvements could be necessary.
“Fiscal year” sounds weird- a better expression?
Do not use “gender” use “sex”.
Response:
Thank you for the comments. We have replaced the words “fiscal year 2017” and “gender” to “April 2017 to March 2018” and “sex”, respectively.
Round 2
Reviewer 1 Report
Thank you. Good job.
Author Response
We appreciate the reviewers' meaningful comments. The additional English proofing was performed. We hope that our manuscript is now suitable for publication.
Reviewer 2 Report
The authors have dealt with the suggestions of improvement in a satisfactory manner. The paper could be accepted.
Author Response

(The authors gave the same response as above.)
